# Characterizing Root Morphological Responses to Exogenous Tryptophan in Soybean (*Glycine max*) Seedlings Using a Scanner-Based Rhizotron System

**DOI:** 10.3390/plants12010186

**Published:** 2023-01-01

**Authors:** Atsushi Sanada, Shinsuke Agehara

**Affiliations:** 1Department of International Agricultural Development, Tokyo University of Agriculture, 1-1-1 Sakuragaoka, Setagaya-ku, Tokyo 156-8502, Japan; 2Gulf Coast Research and Education Center, Institute of Food and Agricultural Sciences, University of Florida, 14625 CR 672, Wimauma, FL 33598, USA

**Keywords:** auxin precursor, biostimulant, image analysis, rhizosphere, root morphology

## Abstract

Tryptophan is a precursor of indole-3-acetic acid (IAA), which is the major auxin involved in the regulation of lateral root formation. In this study, we used a scanner-based rhizotron system to examine root growth and morphological responses of soybean (*Glycine max*, ‘Golden Harvest’) seedlings to exogenous tryptophan. Seeds were sown directly in the rhizotron filled with field soil. Tryptophan was applied at 1.9 and 3.8 mg plant^−1^ by soil drenching or foliar spray. Canopy and root projected area were monitored by analyzing canopy and rhizotron images using ImageJ software. Seedlings were sampled at the first trifoliate stage, 18 days after sowing (DAS), and root morphology was determined by analyzing washed root images using WinRHIZO software. According to contrast analysis, when all tryptophan treatments were pooled, tryptophan application increased canopy and root projected area by 13% to 14% compared with the control at 18 DAS. Tryptophan application also increased root dry matter accumulation by 26%, root:shoot ratio by 24%, and secondary root number by 13%. Tryptophan applied by soil drenching also increased root length and surface area of fine roots (<0.2 mm diameter) by 25% and 21%, respectively, whereas it slightly inhibited primary root elongation. The efficacy of tryptophan soil drenching in stimulating root formation became greater with increasing the application rate. These results suggest that exogenous tryptophan induces auxin-like activities in root development. Soil drenching of tryptophan appears to be an effective strategy in improving the establishment of soybean. Importantly, this strategy is easily implementable by commercial growers with no negative side effect.

## 1. Introduction

Tryptophan is a non-polar aromatic amino acid synthesized in plants [1]. It is also known as a precursor of indole-3-acetic acid (IAA), the most common naturally occurring auxin [1]. IAA promotes cell division and plays an important role in the induction of lateral root formation [2,3,4]. The synthesis of IAA from tryptophan occurs not only in plants but also in rhizosphere microorganisms. In plants, it involves two steps that are mediated by tryptophan aminotransferase and YUC flavin-containing monooxygenases [5,6,7]. By contrast, rhizosphere microorganisms have different IAA synthesis pathways, such as the indole-3-acetamide-mediated pathway by *Pseudomonas syringae* and the indole-3-pyruvic acid-mediated pathway by *Agrobacterium tumefaciens* [8,9,10].

Because tryptophan is a precursor of IAA that promotes root development, growth-promoting effects of tryptophan application have been investigated in several crops. In a greenhouse experiment, Adou Dahab et al. [11] reported that foliar spray application of tryptophan at 100 mg L^−1^ increased plant height by 46%, leaf number by 57%, and leaf area by 34% in blushing philodendron (*Philodendron erubescens*). Sudadi et al. [12] also reported that foliar spray application of tryptophan at 1 mg L^−1^ increased yield of soybean by up to 102% (*Glycine max*) in a greenhouse experiment. In addition, Mustafa et al. [13] found that soil application of tryptophan at 40 mg kg^−1^ increased plant height by 58% and fruit length by 27% in okra (*Abelmoschus esculentus*) in a greenhouse experiment. However, these previous studies evaluated only above-ground growth and yield, and the efficacy of tryptophan as a root growth-promoting agent is still unclear.

Investigation of root morphological traits is challenging because of the laborious process and difficulties in obtaining valid root samples without excessive damage. Traditional root investigation methods include root excavation and soil core sampling [14]. By contrast, recent root research often uses rhizotron systems and image analysis procedures that allow non-destructive and repeated root morphological analysis. Several image processing and analysis programs are available for root measurements, including WinRHIZO Tron, RootSnap, ImageJ, and RootNav 2.0. Recently, Seethepalli et al. [15] introduced RhizoVision Analyzer, which is open-source software developed for high-throughput root crown phenotyping. These new root analysis methods have been employed in many studies. Adu et al. [16] evaluated root morphological traits of *Brassica rapa* genotypes using a scanner-based rhizotron system. Davies et al. [17] also used a scanner-based rhizotron system and evaluated root responses of beech (*Fagus sylvatica*) to indole-3-butyric acid (IBA, a naturally occurring auxin) treatment. Nagel et al. [18] analyzed root geometry and temporal growth responses in Arabidopsis (*Arabidopsis thaliana* L. Heynh) and barley (*Hordeum vulgare*) using an automatic phenotyping system developed for rhizotrons.

We developed a scanner-based rhizotron system suitable for non-destructive root evaluation of soybean seedlings [19]. In this rhizotron system, soybean root projected area showed a significant linear correlation between two image analysis programs, ImageJ and WinRHIZO Tron, throughout the evaluation period [19]. Furthermore, there was a significant correlation in root projected area measured undestructively by ImageJ and destructively by WinRHIZO. ImageJ is an open-source image processing program used in diverse scientific fields. By contrast, WinRHIZO and WinRHIZO Tron are commercial programs developed specifically for root image analysis. WinRHIZO allows automatic and interactive root measurements for washed root images, whereas WinRHIZO Tron performs manual root measurements for rhizotron or in situ root images.

Using the rhizotron system and root image analysis protocols developed in our previous study, the objectives of this study were to characterize root morphological responses of soybean seedlings to exogenous tryptophan and to evaluate its potential as a biostimulant.

## 2. Results

### 2.1. Canopy Projected Area

Overhead canopy images acquired at 18 days after sowing (DAS) are presented in Figure 1. Canopy projected area data are presented in Table 1. In the control, canopy projected area increased steadily by 116% from 11 to 18 DAS (11.7 vs. 25.3 cm^2^ plant^−1^). Both multiple comparisons and contrast analysis detected significant differences between the control and tryptophan treatments only at 18 DAS. The tryptophan soil drench treatment at 1.9 mg plant^−1^ had 27% greater canopy projected area than the control (25.3 vs. 32.2 cm^2^ plant^−1^, *p* < 0.05). The pooled soil drench treatments had 20% greater canopy projected area than the control (25.3 vs. 30.3 cm^2^ plant^−1^, *p* = 0.0218). When pooling all tryptophan treatments, the pooled treatments had 14% greater canopy projected area than the control (25.3 vs. 28.9 cm^2^ plant^−1^, *p* = 0.0672).

Contrast analysis also detected significant differences between the two tryptophan application methods. The pooled soil drench treatments had 19% and 11% greater canopy projected area than the pooled foliar spray treatments at 11 (11.2 vs. 13.3 cm^2^ plant^−1^, *p* = 0.0998) and 18 DAS (27.4 vs. 30.3 cm^2^ plant^−1^, *p* = 0.0985), respectively.

When tryptophan was applied through foliar spray, it had no significant effect on canopy projected area throughout the experiment.

### 2.2. Root Projected Area (Rhizotron Image Analysis)

Rhizotron root images acquired at 18 DAS are presented in Figure 2B. Root projected area data are presented in Table 2. In the control, root projected area increased steadily by 513% from 5 to 18 DAS (1.21 vs. 7.42 cm^2^ plant^−1^). Both multiple comparisons and contrast analysis detected significant differences only at 18 DAS. The tryptophan soil drench treatment at 3.8 mg plant^−1^ had 29% greater root projected area than the control (7.42 vs. 9.60 cm^2^ plant^−1^, *p* < 0.10). The pooled soil drench treatments had 11% greater root projected area than the control (7.42 vs. 8.27 cm^2^ plant^−1^, *p* = 0.0936). The pooled foliar spray treatments had 14% greater root projected area than the control (7.42 vs. 8.47 cm^2^ plant^−1^, *p* = 0.0717). When pooling all tryptophan treatments, the pooled treatments had 13% greater root projected area than the control (7.42 vs. 8.37 cm^2^ plant^−1^, *p* = 0.0670).

No significant difference was detected between soil drench and foliar spray treatments throughout the experiment.

### 2.3. Root Diameter and Lateral Root Number (WinRHIZO Image Analysis)

Seedlings sampled at 18 DAS are shown in Figure 3A, and their washed roots are shown in Figure 3B. Root diameter and lateral root number measured after destructive sampling at 18 DAS are presented in Table 3. The average root diameter ranged from 400 and 424 µm was unaffected by tryptophan treatments.

Secondary root number showed significant treatment effects, according to both multiple comparisons and contrast analysis. The tryptophan soil drench treatment at 1.9 mg plant^−1^ and foliar spray treatments at 1.9 and 3.8 mg plant^−1^ had 13% to 18% greater secondary root number than the control (120 vs. 135–142 plant^−1^, *p* < 0.10). The pooled soil drench treatments had 10% greater secondary root number than the control (120 vs. 132 plant^−1^, *p* = 0.0841). The pooled foliar spray treatments had 16% greater secondary root number than the control (120 vs. 139 plant^−1^, *p* = 0.0066). When pooling across all tryptophan treatments, the pooled treatments had 13% greater secondary root number than the control (120 vs. 136 plant^−1^, *p* = 0.0594).

Tertiary root number ranged from 1475 to 1844 plant^−1^ and was unaffected by tryptophan treatments.

### 2.4. Root Length (WinRHIZO Image Analysis)

Root length data collected after destructive sampling at 18 DAS are presented in Table 4. Responses of primary root length to tryptophan varied between the two application methods. The tryptophan foliar spray treatment at 1.9 mg plant^−1^ had 10% longer primary root length than the control (28.8 vs. 31.7 cm plant^−1^, *p* < 0.10). The tryptophan foliar spray treatments at 1.9 and 3.8 mg plant^−1^ had 15% and 13% longer primary root length than the tryptophan soil drench treatment at 3.8 mg plant^−1^, respectively (27.5 vs. 31.2–31.7 cm plant^−1^, *p* < 0.10). The pooled foliar spray treatments had 9% longer primary root length than the control and the pooled soil drench treatments (28.8–28.9 vs. 31.4 cm plant^−1^, *p* = 0.0384).

Secondary root length (≥0.2 mm diameter) ranged from 497 to 580 cm plant^−1^ and was unaffected by tryptophan treatments.

Tertiary root length (<0.2 mm diameter) showed significant treatment effects, according to contrast analysis. The pooled soil drench treatments had 25% greater tertiary root length than the control and the pooled foliar spray treatments (205 vs. 250 cm plant^−1^, *p* = 0.0428–0.0522). No significant difference was detected between the control and the pooled foliar spray treatments.

The pooled soil drench treatments had 16% greater total lateral root length (secondary + tertiary roots) than the control (702 vs. 812 cm plant^−1^, *p* = 0.0860). Similarly, the pooled soil drench treatments had 16% greater total root length (primary + lateral) than the control (731 vs. 851 cm plant^−1^, *p* = 0.0858). No other significant differences were detected in these root length variables.

### 2.5. Root Surface Area (WinRHIZO Image Analysis)

Root surface area data collected after destructive sampling at 18 DAS are presented in Table 5. According to contrast analysis, primary root surface area was 7% greater in the pooled foliar spray treatments than in the pooled soil drench treatments (13.5 to 14.4 cm^2^ plant^−1^, *p* = 0.0848).

Secondary root surface area (≥0.2 mm diameter) ranged from 66.6 to 80.5 cm^2^ plant^−1^ and was unaffected by tryptophan treatments.

Tertiary root surface area (<0.2 mm diameter) showed significant treatment effects, according to contrast analysis. The pooled soil drench treatments had 21% to 22% greater tertiary root surface area than the control and the pooled foliar spray treatments (9.1–9.2 vs. 11.1 cm^2^ plant^−1^, *p* = 0.0509–0.0770). No significant difference was detected between the control and the pooled foliar spray treatments.

The pooled soil drench treatments had 13% greater total lateral root surface area (secondary + tertiary roots) than the pooled foliar spray treatments (77.8 vs. 88.0 cm^2^ plant^−1^, *p* = 0.0905). No other significant differences were detected.

Total root surface area (primary + lateral roots) was unaffected by tryptophan treatments, according to both multiple comparisons and contrast analysis. There was a significant correlation between root projected area measured non-destructively on rhizotron images with ImageJ software and root surface area measured destructively on washed-root images with WinRHIZO software in soybean seedlings (Figure 4).

### 2.6. Shoot and Root Growth (Plant Sampling)

Shoot ad root growth data collected after destructive plant sampling at 18 DAS are presented in Table 6. Stem diameter showed significant treatment effects, according to multiple comparisons. The tryptophan soil drench treatment at 3.8 mg plant^−1^ had 8% larger stem diameter than the soil drench treatment at 1.9 mg plant^−1^ and the foliar spray treatment at 3.8 mg plant^−1^ (2.45 vs. 2.65 mm, *p* < 0.05).

Leaf area showed significant treatment effects, according to both multiple comparisons and contrast analysis. The tryptophan soil drench treatment at 3.8 mg plant^−1^ had 15% and 30% greater leaf area than the foliar spray treatments at 1.9 and 3.8 mg plant^−1^, respectively (32.0–36.3 vs. 41.6 cm^2^ plant^−1^, *p* < 0.10). The pooled soil drench treatments had 16% greater leaf area than the pooled foliar spray treatments (34.2 vs. 39.8 cm^2^ plant^−1^, *p* = 0.0139).

Shoot dry weight ranged from 183 to 213 g plant^−1^ and was unaffected by tryptophan treatments. By contrast, root dry weight showed significant treatment effects, according to both multiple comparisons and contrast analysis. The tryptophan soil drench treatment at 3.8 mg plant^−1^ and foliar spray treatment at 1.9 mg plant^−1^ had 43% and 25% greater root dry weight than the control, respectively (60.5 vs. 75.7–86.3 g plant^−1^, *p* < 0.05). Increasing the tryptophan application rate for soil drenching from 1.9 to 3.8 mg plant^−1^ increased root dry weight by 19% (72.3 vs. 86.3 g plant^−1^, *p* < 0.05). The pooled soil drench and foliar spray treatments had 31% and 20% greater root dry weight than the control, respectively (60.5 vs. 72.8–79.3 g plant^−1^, *p* = 0.0013–0.0261). When pooling all tryptophan treatments, the pooled treatments had 26% greater root dry weight than the control (60.5 vs. 76.0 g plant^−1^, *p* = 0.0046).

Root:shoot ratio also showed significant treatment effects. According to multiple comparisons, all tryptophan treatments had 20% to 29% greater root:shoot ratio than the control (0.315 vs. 0.379–0.406, *p* < 0.05), and there was no significant difference among tryptophan treatments. The pooled soil drench and foliar spray treatments had 22% and 26% greater root:shoot ratio than the control, respectively (0.315 vs. 0.384–0.397, *p* = 0.0005–0.0027). When pooling all tryptophan treatments, the pooled treatments had 24% greater root:shoot ratio than the control (0.315 vs. 0.390, *p* = 0.0005).

## 3. Discussion

### 3.1. Exogenous Tryptophan Can Induce Auxin-Like Activities in Root Development

Tryptophan is a precursor of IAA, which is a phytohormone playing an important role in lateral root formation [1]. On the other hand, IAA is also involved in the inhibition of primary root growth [20]. In this study, using a scanner-based rhizotron system, root morphological responses of soybean seedlings to exogenous tryptophan were characterized as increases in lateral root formation, root dry matter accumulation, and root:shoot ratio. Furthermore, tryptophan applied by soil drenching inhibited primary root elongation when increasing the application rate from 1.9 to 3.8 mg plant^−1^. These results suggest that exogenous tryptophan can induce auxin-like activities in root development.

The mechanism of IAA-induced lateral root formation involves the transportation of IAA from root tips to lateral root prebranch sites, followed by the formation of lateral root founder cells [21]. Casimiro et al. [4] reported that Arabidopsis treated with N-1-naphthylphthalamic acid, a polar auxin transport inhibitor, reduced IAA transportation from root tips, thereby inhibiting lateral root formation. In plants, IAA synthesis from tryptophan occurs through a two-step pathway regulated by TAA and YUC enzymes [5,6,7]. These enzymes are generally abundant in various plant tissues, including leaves, stems, and roots [22]. In this study, tryptophan promoted lateral root formation whether it was applied by soil drenching or foliar spray, suggesting that tryptophan absorbed in plant tissues can be transformed readily into IAA and transported to prebranch sites.

This study is the first to characterize root morphological responses to exogenous tryptophan using root image analysis. Root morphological modification could be a contributing factor to the previously reported beneficial effects of exogenous tryptophan on plant growth and drought stress tolerance [11,12,13,23,24,25].

### 3.2. Tryptophan Is More Effective in Promoting Fine Root Development When Applied by Soil Drenching Than by Foliar Spray

Although growth promoting effects of exogenous tryptophan have been reported for both soil drenching and foliar spray treatments [11,13,25], comparison of root responses to the two methods has not been performed in the same study. Long-distance transport of amino acids from leaves to other tissues is facilitated via phloem [26]. Amino acids in mesophyll cells are transported to phloem through symplastic or apoplastic pathways and loaded into phloem by various amino acid transporters [26]. Therefore, even when tryptophan is applied by foliar spray, it may exert its effects in roots through translocation. In this study, although both tryptophan application methods promoted secondary root formation, only the soil drenching method showed efficacy in promoting tertiary root development (<0.2 mm in diameter). This finding implies that tryptophan needs to be applied to roots to fully exert its effects on fine root formation. The importance of localized supply is also reported for auxin-induced lateral root formation [27,28].

The lack of efficacy of tryptophan spray application in promoting fine root development may be associated with the limited supply of tryptophan to prebranch sites. First, transport of absorbed tryptophan from leaves to roots is limited by the number of tryptophan transporters and the availability of cellular energy [29]. Second, because tryptophan can be metabolized into not only auxins but also other compounds such as glucosinolates, phytoalexins, and alkaloids, the rate of tryptophan transport to prebranch sites is also determined by the metabolic pathway [30]. Third, even if absorbed tryptophan is converted to IAA, IAA can be inactivated through catabolism before reaching prebranch sites [31].

### 3.3. The Optimum Application Rate of Tryptophan Depends on the Application Method

Concentration-dependent effects of exogenous tryptophan on plant growth have been reported for both soil drenching and foliar spray application [11,13,25]. In a review article, Mustafa et al. [24] reported that the optimum tryptophan concentration for soil drenching ranged from 0.0001 to 100 mg kg^−1^ soil in 16 publications, whereas that for foliar spray application ranged from 1 to 350 mg L^−1^ in 12 publications. In this study, tryptophan was applied at 1.9 or 3.8 mg plant^−1^ by both methods but in different concentrations and volumes: 19 or 38 mg L^−1^ at 100 mL plant^−1^ for soil drenching and 1900 or 3800 mg L^−1^ at 1 mL plant^−1^ for foliar spray. For soil drench treatments, tryptophan application rates of 1.9 and 3.8 mg plant^−1^ corresponded to 0.43 or 0.86 mg kg^−1^ soil. Therefore, compared to the literature, the tryptophan concentrations used for soil drenching were comparable, but those for foliar spray were relatively high.

In this study, when tryptophan was applied through foliar spray, both tryptophan concentrations promoted root growth to a similar extent, suggesting that the effectiveness of tryptophan spray application could have been maximized at or below a concentration of 1900 mg L^−1^. Tryptophan is the main precursor of IAA in plants [32]. The conversion of tryptophan to IAA is facilitated by TAA and YUC enzymes [5,6,7]. For IAA to exert its effects, IAA must bind to auxin receptors and promote auxin response factors [1]. Therefore, the abundance of TAA and YUC enzymes and auxin receptors may be a limiting factor for IAA signaling and thus tryptophan-induced growth promotion.

By contrast, growth promotion by tryptophan soil drenching became more pronounced with increasing tryptophan concentration from 19 to 38 mg L^−1^, suggesting that growth-promoting effects of tryptophan soil drenching could be enhanced further with a concentration above 38 mg L^−1^ (0.86 mg kg^−1^ soil). These results collectively demonstrate that the optimum application rate of tryptophan depends on the treatment method.

### 3.4. Practical Implementation

The fact that even a single application of tryptophan significantly improved soybean seedling growth demonstrates its high efficacy as a biostimulant. Tryptophan has important features for successful commercial implementation. First, its application is easy and flexible, as it can be applied through soil drenching or foliar spray. Second, it can exert significant growth-promoting effects at a low application rate, 1.9 to 3.8 mg plant^−1^. Third, it may be approved for organic agriculture because it is produced through fermentation, according to the manufacturer.

Interestingly, one of the most consistent effects of tryptophan was increased dry matter partitioning to roots. This growth modification may enhance plant tolerance to abiotic stress, such as water stress and nutrient deficiency [33,34,35]. Further research on how exogenous tryptophan affects abiotic stress tolerance is needed to better understand its full potential as a biostimulant.

## 4. Materials and Methods

### 4.1. Tryptophan Treatments

There were five treatments: water control, soil drench application of tryptophan at 1.9 and 3.8 mg plant^−1^, and foliar spray application of tryptophan at 1.9 and 3.8 mg plant^−1^. Tryptophan was applied at 1.9 and 3.8 mg plant^−1^ by both methods but with different concentrations and volumes: 19 and 38 mg L^−1^ at 100 mL plant^−1^ for soil drenching and 1900 and 3800 mg L^−1^ at 1 mL plant^−1^ for foliar spray. Soil drench treatments were performed immediately after emergence (5 DAS), whereas foliar spray treatments were performed at the unifoliate leaf stage (10 DAS). Deionized water was used to prepare all tryptophan solutions. The control plants were sprayed with deionized water (1 mL plant^−1^) at 10 DAS. All treatments were performed between 10:00 and 11:00 a.m.

### 4.2. Rhizotron Experiment

A greenhouse experiment was conducted at the University of Florida’s Gulf Coast Research and Education Center in Balm, Florida, United States (latitude 27°76′, longitude 82°23′W; elevation 39 m) using a scanner-based rhizotron system described by Agehara and Sanada [19]. The soil at this study site is classified as Myakka fine sand (sandy, siliceous, Hyper-thermic Oxyaquic Alorthods). Granular fertilizers were incorporated into the soil at a depth of 15 cm to supply 56, 49, and 93 kg of nitrogen, phosphorous, and potassium per hectare, respectively. The surface (15 cm depth) soil was collected and sieved through 5 mm mesh to remove large organic residues and other debris. The soil had pH of 6.7 and organic matter content of 10 g kg^−1^. Each rhizotron was packed with the soil at 2655 cm^3^ using the same bulk density as in the field (1.67 g cm^–3^). Rhizotrons were inclined at 30° on a rack to maximize the root contact on the lower scanning window (Figure 5). Three seeds were sown 2 cm deep along the lower scanning window in each rhizotron. After emergence, two seedlings were thinned to keep only one uniform plant per rhizotron. Plants were grown in rhizotrons until 18 DAS.

### 4.3. Canopy Image Analysis

Overhead canopy images were acquired using a digital camera (Cyber-shot DSC-RX100; Sony, Tokyo, Japan) at about 1 m above the plant canopy at 11, 14, and 18 DAS. All images were saved in JPEG format. Images were processed using ImageJ software (http://rsb.info.nih.gov/ij/, accessed on 4 August 2022) following the methods described by Agehara [36].

### 4.4. Rhizotron Root Image Analysis

Rhizotron root images were acquired using a flat-bed scanner (perfection V800; Epson, Nagano, Japan) at 5, 8, 11, 14, and 18 DAS. All images were saved in JPEG format at 300 dots per inch (dpi). Root scanning was performed only on the lower rhizotron window throughout the experiment because no roots were visible on the upper rhizotron window.

Root images were processed and analyzed using ImageJ software. After converting images from 24-bit color to 8-bit grayscale, thresholding was performed to distinguish root pixels from all background pixels. The optimal threshold values were selected to remove background pixels with minimum changes in root diameter. After thresholding, images were converted to a binary format in which root and background pixels were displayed in black and white, respectively. The total area of black pixels was measured and recorded as root projected area.

### 4.5. Plant Sampling and Growth Measurements

At the end of the experiment (18 DAS), plants were sampled from rhizotrons by gently washing roots to remove soil. Stem diameter was measured immediately below the cotyledonary node using a digital caliper (Absolute Digimatic Caliper Series 500; Mitutoyo, Kanagawa, Japan). Leaf area was measured using an optical area meter (LI-3100; LI-COR, Lincoln, NE, USA). Roots were separated from the shoot at the root–shoot junction. Shoots were dried at 65 °C for 48 h to determine dry weight. Roots were washed thoroughly and stored in 60% ethanol solution until root scanning. After root scanning, roots were dried at 65 °C for 48 h to determine dry weight. Root scanning and image analysis procedures are described below.

### 4.6. Image Analysis of Washed Roots

Washed roots were placed in an acrylic tray filled with water and carefully spread to minimize the overlapping of roots. Roots were then scanned using a flat-bed scanner (EPSON 10000 XL, Epson, Nagano, Japan). All images were saved in JPEG format at 400 dpi. Root images were analyzed using WinRHIZO software (WinRHIZO Arabidopsis 2019a; Regent Instruments, Quebec, Canada). Root images were first converted into binary images using the threshold setting that optimized the segmentation of roots. Images were then analyzed to determine several root traits, including root diameter, number of tips (root number), root length, and root surface area. Two diameter classes were used for >0.2 mm and ≤0.2 mm.

Primary roots were visually determined, and primary root length and surface area were measured using SmartRoot, a plugin for ImageJ. Secondary root length and surface area were calculated by subtracting the primary root data from the WinRHIZO data of roots with a thickness of >0.2 mm. Tertiary root length and surface area were determined by WinRHIZO with a diameter threshold of ≤0.2 mm.

### 4.7. Experiment Design and Statistical Analysis

Treatments were arranged in a randomized complete block design. Each treatment had six replicates (rhizotrons).

All data were analyzed using the generalized linear mixed model procedure (PROC GLIMMIX) in SAS statistical software (SAS 9.4; SAS Institute, Cary, NC, USA). The best model was selected based on the smallest corrected Akaike information criterion. Continuous data (all data except for root number) were modeled with the lognormal distribution (DIST = LOGNORMAL). For model parameter estimation, boundary constraints on covariance were removed (NOBOUND), and degrees of freedom for the fixed effects were adjusted by the Kenward–Roger degrees of freedom approximation (DDFM = KR). Count data (root number) were modeled with the negative binomial distribution (DIST = NEGBIN). Model parameters were estimated by using maximum likelihood estimation with quadrature approximation (METHOD = QUAD) and default bias-corrected sandwich estimators (EMPIRICAL = MBN)

Multiple comparisons of least squares means were performed using the Fisher’s LSD method. In addition, contrast analysis was used to test specific hypotheses. First, we hypothesized that all tryptophan treatments have equivalent growth modulating effects, thereby comparing the control with the pooled tryptophan treatments. Second, we hypothesized that tryptophan has different growth modulating effects depending on the application method, thereby comparing the control with the pooled soil drench treatments or with the pooled foliar spray treatments. We also compared the pooled soil drench treatments and the pooled foliar spray treatments. For all data analyzed, *p* values less than 0.05 or 0.10 were considered to be statistically significant.

Continuous data were back-transformed by exponentiating the sum of the least square mean and the correction factor. Count data were rescaled to the original scale by using the inverse link option (ILINK) in the LSMEANS statement. Back-transformed or rescaled data are reported in this study.

## Figures and Tables

**Figure 1 plants-12-00186-f001:**
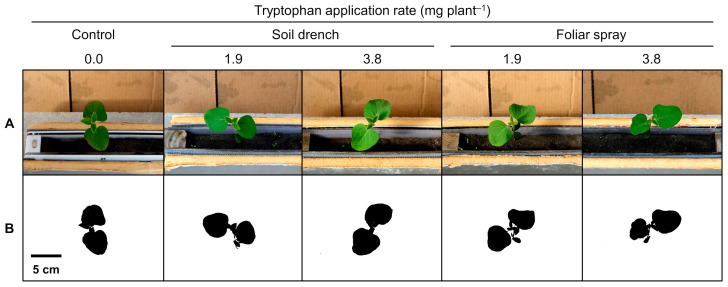
Overhead canopy images of soybean seedlings at 18 days after sowing: (**A**) color images; (**B**) binary images processed by ImageJ software. Tryptophan soil drench and foliar spray treatments were performed at 5 and 10 days after sowing (immediately after emergence and at the unifoliate leaf stage), respectively. The application volumes were 100 mL plant^−1^ for soil drench treatments and 1 mL plant^−1^ for foliar spray treatments. The control plants were sprayed with water (1 mL plant^−1^) at 10 days after sowing.

**Figure 2 plants-12-00186-f002:**
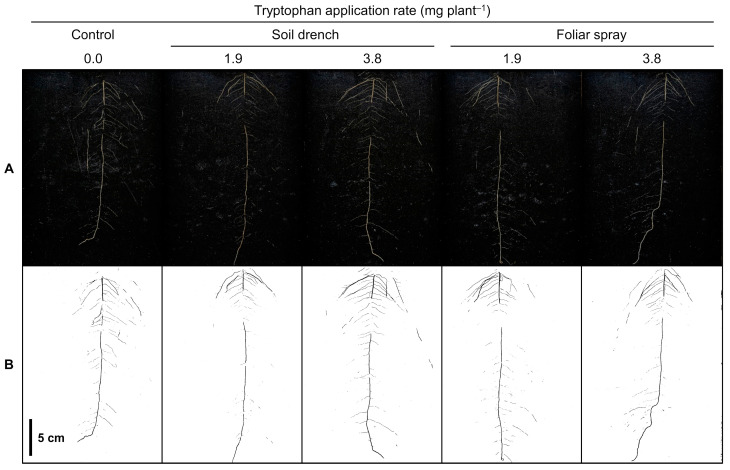
Rhizotron root images of soybean seedlings at 18 days after sowing: (**A**) color images; (**B**) binary images processed by ImageJ software. Tryptophan treatments were as described in Figure 1.

**Figure 3 plants-12-00186-f003:**
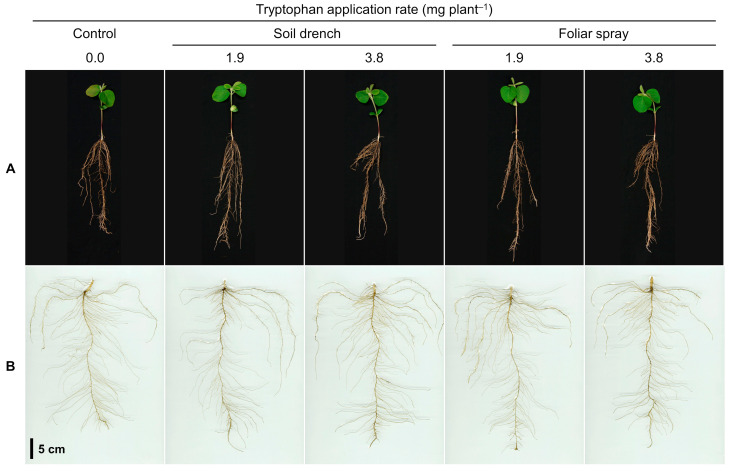
Soybean seedlings sampled at 18 days after sowing: (**A**) photographs; (**B**) scan images of washed roots. Tryptophan treatments were as described in Figure 1.

**Figure 4 plants-12-00186-f004:**
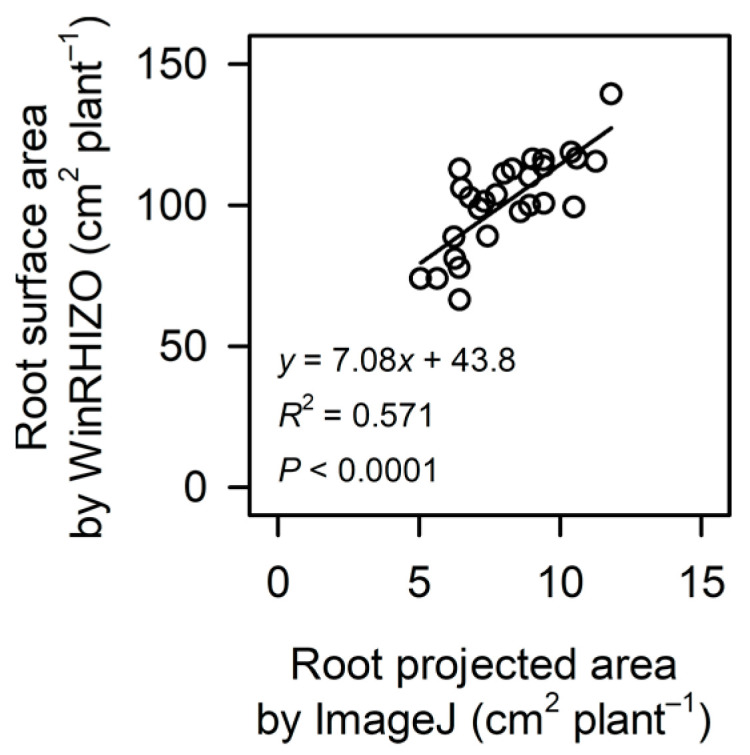
A linear correlation between root projected area measured non-destructively on rhizotron images with ImageJ software and root surface area measured destructively on washed-root images with WinRHIZO software in soybean seedlings. All images were collected at 18 days after sowing. Treatments were as described in Table 1.

**Figure 5 plants-12-00186-f005:**
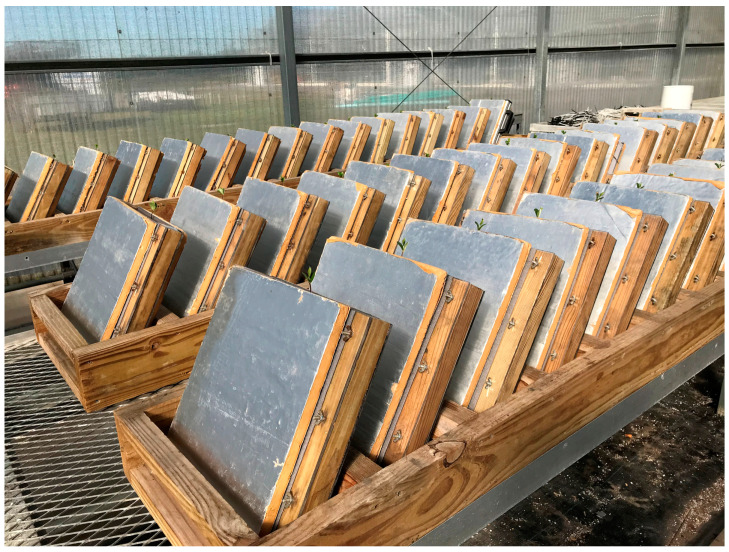
Soybean seedlings grown in rhizotrons.

**Table 1 plants-12-00186-t001:** Canopy projected area of soybean seedlings as affected by tryptophan (TP) soil drench and foliar spray treatments ^1^.

Treatment	TP Application Rate	Canopy Projected Area (cm^2^ plant^−1^)		
Method ^2^	(mg plant^−1^)	11 DAS	14 DAS	18 DAS
Control	0.0	11.7	21.7	25.3 ^b^
Soil drench	1.9	11.7	21.9	28.5 ^ab^
	3.8	15.0	26.7	32.2 ^a^
Foliar spray	1.9	11.3	22.7	29.3 ^ab^
	3.8	11.2	20.8	25.5 ^b^
		Pooled data		
All TP treatments		12.3	23.0	28.9
All TP soil drench treatments		13.3	24.3	30.3
All TP foliar spray treatments		11.2	21.8	27.4
		*p* value		
Treatment effect		0.1929	0.1508	0.0440
Untreated vs. All TP		0.6606	0.4930	0.0672
Untreated vs. All TP soil drench		0.2796	0.2226	0.0218
Untreated vs. All TP foliar spray		0.7691	0.9873	0.3056
All TP soil drench vs. All TP foliar spray		0.0998	0.1436	0.0985

^1^ Means followed in a column followed by the same lowercase letter are not significantly different at *p* < 0.05 (Tukey–Kramer test). Contrast analysis was used to test specific hypotheses. ^2^ Tryptophan treatments were as described in Figure 1.

**Table 2 plants-12-00186-t002:** Root projected area of soybean seedlings as affected by tryptophan (TP) soil drench and foliar spray treatments ^1^.

Treatment	TP Application Rate	Root Projected Area (cm^2^ plant^−1^)				
Method ^2^	(mg plant^−1^)	5 DAS	8 DAS	11 DAS	14 DAS	18 DAS
Control	0.0	1.21	3.47	5.15	6.27	7.42 ^B^
Soil drench	1.9	0.99	2.87	4.25	5.61	6.93 ^B^
	3.8	1.14	3.78	5.84	7.35	9.60 ^A^
Foliar spray	1.9	1.12	3.67	5.34	6.85	8.36 ^AB^
	3.8	1.10	3.56	5.60	6.75	8.58 ^AB^
		Pooled data				
All TP treatments		1.09	3.47	5.26	6.64	8.37
All TP soil drench treatments		1.07	3.32	5.05	6.48	8.27
All TP foliar spray treatments		1.11	3.62	5.47	6.80	8.47
		*p* value				
Treatment effect		--	0.3371	0.3601	0.3863	0.0670
Untreated vs. All TP		--	0.3725	0.3754	0.2474	0.0570
Untreated vs. All TP soil drench		--	0.6593	0.5984	0.3719	0.0936
Untreated vs. All TP foliar spray		--	0.2383	0.2778	0.2256	0.0717
All TP soil drench vs. All TP foliar spray		--	0.3871	0.5152	0.7176	0.9108

^1^ Means followed in a column followed by the same uppercase letter are not significantly different at *p* < 0.10 (Tukey–Kramer test). Contrast analysis was used to test specific hypotheses. ^2^ Tryptophan treatments were as described in Figure 1.

**Table 3 plants-12-00186-t003:** Root diameter and lateral root number of soybean seedlings as affected by tryptophan (TP) soil drench and foliar spray treatments ^1^.

Treatment	TP Application Rate	Root Diameter ^3^	Root Number per Plant	
Method ^2^	(mg plant^−1^)	(µm)	Secondary Roots	Tertiary Roots
Control	0.0	425	120 ^C^	1590
Soil drench	1.9	400	135 ^AB^	1786
	3.8	421	128 ^BC^	1844
Foliar spray	1.9	412	142 ^A^	1591
	3.8	410	137 ^AB^	1475
		Pooled data		
All TP treatments		411	136	1674
All TP soil drench treatments		410	132	1815
All TP foliar spray treatments		411	139	1533
		*p* value		
Treatment effect		0.3571	0.0594	0.9583
Control vs. All TP		0.1591	0.0144	0.9386
Control vs. All TP soil drench		0.2051	0.0841	0.7546
Control vs. All TP foliar spray		0.1965	0.0066	0.7524
All TP soil drench vs. All TP foliar spray		0.9948	0.1822	0.5371

^1^ Data were collected at 18 days after sowing. Means followed in a column followed by the same uppercase letter are not significantly different at *p* < 0.10 (Tukey–Kramer test). Contrast analysis was used to test specific hypotheses. ^2^ Tryptophan treatments were as described in Figure 1. ^3^ The average root diameter was calculated using all roots detected by WinRHIZO software.

**Table 4 plants-12-00186-t004:** Root length of soybean seedlings as affected by tryptophan (TP) soil drench and foliar spray treatments ^1^.

		Root Length (cm plant^−1^)				
Treatment	TP Application Rate	Primary	Lateral Roots ^3^			Primary +
Method ^2^	(mg plant^−1^)	Root	≥0.2 mm	<0.2 mm	Total	Lateral
Control	0.0	28.8 ^BC^	497	205	702	731
Soil drench	1.9	30.2 ^AB^	547	251	798	828
	3.8	27.5 ^C^	580	266	847	874
Foliar spray	1.9	31.7 ^A^	523	222	745	777
	3.8	31.2 ^AB^	505	202	707	738
		Pooled data				
All TP treatments		30.2	527	231	758	804
All TP soil drench treatments		28.9	554	257	812	851
All TP foliar spray treatments		31.4	499	205	704	757
		*p* value				
Treatment effect (ANOVA)		0.0518	0.5605	0.1934	0.3423	0.3554
Untreated vs. All TP		0.0221	0.3177	0.2148	0.2422	0.2337
Untreated vs. All TP soil drench		0.9276	0.1589	0.0522	0.0860	0.0858
Untreated vs. All TP foliar spray		0.0384	0.7061	0.8348	0.7192	0.6895
All TP soil drench vs. All TP foliar spray		0.0228	0.2150	0.0428	0.1037	0.1128

^1^ Data were collected at 18 days after sowing. Contrast analysis was used to test specific hypotheses. ^2^ Tryptophan treatments were as described in Figure 1. ^3^ Lateral roots were sorted in two diameter classes. Roots ≥ 2 mm and < 2 mm diameter represent secondary and tertiary roots, respectively.

**Table 5 plants-12-00186-t005:** Root surface area of soybean seedlings as affected by tryptophan (TP) soil drench and foliar spray treatments ^1^.

		Root Surface Area (cm^2^ plant^−1^)				
Treatment	TP Application Rate	Primary	Lateral Roots ^3^			Primary +
Method ^2^	(mg plant^−1^)	Roots	≥0.2 mm	<0.2 mm	Total	Lateral
Control	0.0	14.4	67.9	9.1	77.0	91.4
Soil drench	1.9	13.6	73.3	10.7	84.0	97.6
	3.8	13.4	80.5	11.6	92.1	105.5
Foliar spray	1.9	14.3	70.5	9.6	80.2	94.5
	3.8	14.6	66.6	8.7	75.4	90.0
		Pooled data				
All TP treatments		14.0	72.8	10.2	82.9	96.9
All TP soil drench treatments		13.5	76.9	11.1	88.0	101.5
All TP foliar spray treatments		14.4	68.6	9.2	77.8	92.2
		*p* value				
Treatment effect		0.4372	0.3626	0.2279	0.3006	0.4072
Untreated vs. All TP		0.4522	0.3814	0.2857	0.3438	0.3941
Untreated vs. All TP soil drench		0.1607	0.1497	0.0770	0.1198	0.1643
Untreated vs. All TP foliar spray		0.9371	0.9120	0.9370	0.9062	0.9034
All TP soil drench vs. All TP foliar spray		0.0848	0.1172	0.0509	0.0905	0.1344

^1^ Data were collected at 18 days after sowing. Contrast analysis was used to test specific hypotheses. ^2^ Tryptophan treatments were as described in Figure 1. ^3^ Lateral roots were sorted in two diameter classes. Roots ≥ 2 mm and < 2 mm diameter represent secondary and tertiary roots, respectively.

**Table 6 plants-12-00186-t006:** Shoot and root growth of soybean seedlings as affected by tryptophan (TP) soil drench and foliar spray treatments ^1^.

Treatment	TP Application Rate	Stem Diameter	Leaf Area	Dry wt (mg plant^−1^)		Root:Shoot
Method ^2^	(mg plant^−1^)	(mm)	(cm^2^ plant^−1^)	Shoot	Roots	Ratio (wt/wt)
Control	0.0	2.56 ^ab^	36.9 ^ABC^	194	60.5 ^c^	0.315 ^b^
Soil drench	1.9	2.40 ^c^	38.1 ^AB^	188	72.3 ^bc^	0.389 ^a^
	3.8	2.65 ^a^	41.6 ^A^	213	86.3 ^a^	0.406 ^a^
Foliar spray	1.9	2.60 ^ab^	36.3 ^BC^	195	75.7 ^ab^	0.388 ^a^
	3.8	2.45 ^bc^	32.0 ^C^	183	69.8 ^bc^	0.379 ^a^
		Pooled data				
All TP treatments		2.52	37.0	195	76.0	0.390
All TP soil drench treatments		2.52	39.8	201	79.3	0.397
All TP foliar spray treatments		2.52	34.2	189	72.8	0.384
		*p* value				
Treatment effect		0.0132	0.0597	0.2796	0.0046	0.0085
Untreated vs. All TP		0.5731	0.9659	0.9850	0.0030	0.0005
Untreated vs. All TP soil drench		0.6162	0.2734	0.6071	0.0013	0.0005
Untreated vs. All TP foliar spray		0.5974	0.3078	0.6311	0.0261	0.0027
All TP soil drench vs. All TP foliar spray		0.9734	0.0139	0.2290	0.1322	0.4287

^1^ Data were collected at 18 days after sowing. Means followed in a column followed by the same lowercase and uppercase letter are not significantly different at *p* < 0.05 and 0.10, respectively (Tukey–Kramer test). Contrast analysis was used to test specific hypotheses. ^2^ Tryptophan treatments were as described in Figure 1.

## Data Availability

The data presented in this study are available on request from the corresponding author.

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
