# Peer review of "Characterizing Root Morphological Responses to Exogenous Tryptophan in Soybean (Glycine max) Seedlings Using a Scanner-Based Rhizotron System"

_plants, 2023, doi:10.3390/plants12010186_

Round 1

Reviewer 1 Report

Novelty of the research article is well justified. It is acceptable for publication in Plants, however, following minor improvements are suggested;

1.      Effect of different rates and modes of L-tryptophan on the No. of tertiary roots per plant of soybean has been shown in the Table No. 3 but there is no explanation about it in the results. Suggest to incorporate brief explanation. (Table 3, Page No. 5)

2.      There is repetition of values “702 vs. 812 cm plant-1, p=0.0860” on Page No. 6. Correction may please be done. (Line no. 172)

3.      In results section, although results are well clear, yet all the subheadings have been explained in single approach which appears like copying the same text in subheadings and changing the values. If feasible, make some changes in elaboration approach.

4.      In discussion section, the link is missing between the current research findings and previous studies regarding effect of exogenous application of auxin or auxin precursor on the root growth of plants. Suggest to incorporate the relevant information, if available.

Author Response

We truly appreciate your constructive comments and suggestions! And thank you so much for your time and the opportunity to revise the paper!

Below we have listed my individual responses to your comments, describing how each of your concern and comment was addressed in the revision.

  1. Effect of different rates and modes of L-tryptophan on the No. of tertiary roots per plant of soybean has been shown in the Table No. 3 but there is no explanation about it in the results. Suggest to incorporate brief explanation. (Table 3, Page No. 5)

Thank you so much for the catch! We added the description for the tertiary root number results. Please see L147-148.

  1. There is repetition of values “702 vs. 812 cm plant-1, p=0.0860” on Page No. 6. Correction may please be done. (Line no. 172)

Thank you so much for the catch again! We deleted the repetition. Please see L178.

  1. In results section, although results are well clear, yet all the subheadings have been explained in single approach which appears like copying the same text in subheadings and changing the values. If feasible, make some changes in elaboration approach.

Thank you for the suggestion. Yes, we agree that all our results are explained in the same format, but we decided to do so mainly to improve the clarity and readability.

  1. In discussion section, the link is missing between the current research findings and previous studies regarding effect of exogenous application of auxin or auxin precursor on the root growth of plants. Suggest to incorporate the relevant information, if available.

We added a new paragraph as suggested. Please see L276-279. We also have discussions referring to the differences between previous and current findings in terms of the effects of exogenous tryptophan in Discussion 3.2 and 3.3.

Reviewer 2 Report

The study presented seems to me clear, simple, but one has to be careful with the results, where some data are not entirely clear and discussed in a somewhat confusing way. I would discuss the control and then compare it with the treatments, first for the soil treatment and then for the foliar treatment, so that the reader has no difficulty in finding the results in the table. The references to the figures in lines 112, 132 and 133 need to be revised.  

Author Response

Thank you so much for reviewing our paper! We truly appreciate all the constructive comments and suggestions!

Below we have listed my individual responses to your comments, describing how each of your concern and comment was addressed in the revision.

The study presented seems to me clear, simple, but one has to be careful with the results, where some data are not entirely clear and discussed in a somewhat confusing way. I would discuss the control and then compare it with the treatments, first for the soil treatment and then for the foliar treatment, so that the reader has no difficulty in finding the results in the table. The references to the figures in lines 112, 132 and 133 need to be revised. 

We added the description for the tertiary root number results. Please see L147-148.

Whenever we observed significant differences between the control and tryptophan treatments, we described the control data, and then described the results of mean comparisons followed by contrast analysis. When there was no significant treatment effect, we just mentioned that treatments had no significant effect.

We corrected the references to the figures as suggested. Thank you so much for the catch!

Reviewer 3 Report

Soybean root research is undoubtedly a very important part.

 - Line 67 to 70

I wanted to read authors previous research, but I couldn't find the paper.

In this paper, there is no information on how exactly the values ​​actually obtained and the values ​​obtained from the program match.

There is no information on how much the measured value actually matches the value obtained from the program. This is information that must be included.

 In my opinion, sampling the roots is going to be hard, but wouldn't it be better to experiment with the v2 stage and beyond? I have confirmed that soybeans are symbiotic when root nodules form at approximately v2 stage.

This is because I think that root nodules have a great influence on the growth characteristics of beans.

Author Response

Thank you so much for reviewing the manuscript and for proving constructive comments and suggestions!

Below I have listed my individual responses to your comments, describing how each of your concern and comment was addressed in the revision.

  1. Line 67 to 70 – I wanted to read authors previous research, but I couldn't find the paper.

Our previous paper is listed in the reference section. The paper was written in Japanese, but the abstract and all figures in the paper were prepared in English.

Agehara, S.; Sanada, A. Quick and inexpensive root and shoot evaluation methods using a scanner-based rhizotron system and ImageJ in soybean (Glycine max). Root Research 2020, 29, 5-19, doi:10.3117/rootres.29.5.

  1. In this paper, there is no information on how exactly the values actually obtained and the values obtained from the program match.

Some data, such as canopy and root projected area, were non-destructive data, so the actual measurements were not performed. However, we provided references (reference #19 and 36) to show the validity of the image analysis procedures used in this study. We also wish to obtain data by both image analysis and manual measurements, but manual measurements are not practical for some root data, such as root surface area and length of fine roots. We adjusted the thresholding values to maximize the accuracy of such data, which are explained in the method section.  

  1. There is no information on how much the measured value actually matches the value obtained from the program. This is information that must be included.

Please see the reply to the comment 2 above.

  1. In my opinion, sampling the roots is going to be hard, but wouldn't it be better to experiment with the v2 stage and beyond? I have confirmed that soybeans are symbiotic when root nodules form at approximately v2 stage. This is because I think that root nodules have a great influence on the growth characteristics of beans.

Thank you so much for the comment! In this experiment, we focused on the initial root development to evaluate the effects of tryptophan without the presence of root nodules. Also, the rhizotron system used in this study is suitable only for the evaluation of initial growth. However, we think it would be an interesting study to look at the effects of tryptophan on soybean roots with root nodules. Hope we can design such an experiment in the future!

Round 2

Reviewer 3 Report

I have confirmed that the authors have kindly responded to those comments.

In the filed of Phenmomics, I strongly insist that some of the total materials must compare between actual and image measurements. In particular, it is essential to use a designed program methods

And I could only check the abstract of the paper that the authors published and I don't think the value of r2 mentioned there is enough to published in this journal.